# Dynamic Hip Screw versus Cannulated Cancellous Screw in Pauwels Type II or Type III Femoral Neck Fracture: A Systematic Review and Meta-Analysis

**DOI:** 10.3390/jpm11101017

**Published:** 2021-10-11

**Authors:** Eic Ju Lim, Hyun-Chul Shon, Jae-Woo Cho, Jong-Keon Oh, Junhyun Kim, Chul-Ho Kim

**Affiliations:** 1Department of Orthopaedic Surgery, Chungbuk National University Hospital, Chungbuk National University College of Medicine, Cheongju 28644, Korea; limeicju@gmail.com (E.J.L.); hyunchuls@chungbuk.ac.kr (H.-C.S.); 2Department of Orthopedic Surgery, Korea University Guro Hospital, Seoul 08308, Korea; jaewoocho@korea.ac.kr (J.-W.C.); jkoh@korea.ac.kr (J.-K.O.); 3Department of Orthopedic Surgery, Gachon University Gil Medical Center, Incheon 21565, Korea; junhyun0761@gmail.com; 4Department of Orthopaedic Surgery, Chung-Ang University Hospital, Chung-Ang University College of Medicine, Seoul 06973, Korea

**Keywords:** femoral neck fracture, Pauwels type, vertical, dynamic hip screw, cannulated screw

## Abstract

Vertically oriented femoral neck fractures (FNFs) are known to be especially unstable FNFs, and they have a higher associated risk of failure. The dynamic hip screw (DHS) technique and the cannulated cancellous screw (CCS) technique are the two main fixation techniques used in the treatment of FNFs. However, no large clinical study has compared the DHS and CCS techniques in patients with high-shear-angle FNFs. MEDLINE, Embase, Cochrane Library, and Web of Science were systematically searched for studies that compared the DHS and CCS techniques for the treatment of Pauwels type II or type III FNF. Pooled analysis was performed to identify differences between the DHS and CCS techniques in Pauwels type II or type III FNF, with a focus on postoperative complications such as fracture nonunion and osteonecrosis of the femoral head (ONFH). We included five studies with a total of 252 patients. The DHS technique was used in 96 patients (DHS group), and the CCS technique was used in 156 patients (CCS group). The pooled analysis revealed that the nonunion rate in the CCS group was significantly higher than that in the DHS group (OR = 0.32; 95% CI, 0.11–0.96; *p* = 0.04, I^2^ = 0%), but there was no difference in the incidence of ONFH between the groups (OR = 0.98; 95% CI, 0.20–4.73; *p* = 0.98, I^2^ = 53%). For vertically oriented FNFs, the DHS technique is more favorable and has a lower risk of fracture nonunion than the CCS technique.

## 1. Introduction

Femoral neck fracture (FNF) is a common injury in orthopedics that has remained unresolved. In terms of economic burden, hip fracture is one of the 20 most expensive diagnoses in the United States, with approximately 20 billion dollars spent on its management, and it is estimated that there will be approximately 300,000 cases of hip fractures annually in the United States by the year 2030 [1].

Several classification systems for the proper treatment of FNF have been introduced, and they include the Pauwels, Garden, and AO/OTA (Arbeitsgemeinschaft für Osteosynthesefragen/Orthopaedic Trauma Association) classification systems [2,3,4]. In the Pauwels classification, FNFs are categorized into three grades according to fracture orientation based on degree of verticality. Pauwels grade I FNFs have fracture angles <30°, grade II FNFs have fracture angles between 30° and 50°, and grade III FNFs have fracture angles >50°. In this grading system, fractures with vertically oriented fracture lines are considered more unstable and have a higher associated risk of failure than horizontal fractures as they are affected by a greater shearing force [5].

The dynamic hip screw (DHS) technique, which uses a fixed-angle device, and the cannulated cancellous screw (CCS) technique are the two main fixation techniques for FNFs. Several biomechanical studies compared the DHS and CCS techniques for vertically oriented FNFs (i.e., Pauwels types II and III FNFs) and reported a higher fixation strength with the DHS technique than with the CCS technique. This supports the notion that fractures with a higher “shear angle” are more unstable and therefore have a higher rate of nonunion or other complications such as osteonecrosis of the femoral head (ONFH) [6,7]. However, to the best of our knowledge, no large clinical studies have compared the DHS and CCS techniques in patients with high-shear-angle FNFs.

Therefore, in this meta-analysis, we aimed to compare the DHS and CCS fixation techniques in Pauwels type II or type III FNFs, with a special focus on nonunion rate and incidence of postoperative ONFH.

## 2. Materials and Methods

This study was conducted in accordance with the guidelines of the Revised Assessment of Multiple Systematic Reviews (R-AMSTAR) and Preferred Reporting Items for Systematic Reviews and Meta-Analyses (PRISMA) [8,9].

### 2.1. Literature Search

In compliance with the referenced guidelines, MEDLINE, Embase, Cochrane Library, and Web of Science were searched for studies that compared DHS and CCS fixation techniques in the treatment of Pauwels type II or type III FNF. Articles published up to 19 August 2020 were identified using an a priori search strategy. Search terms included synonyms and related terms for FNF, DHS, and CCS as follows: ((“Fracture*”) AND (“femur neck” OR “femoral neck” OR “intracapsular”)) AND (dynamic OR compression OR “fixed angle” OR screw* OR device*) AND ((cannulated OR cancellous) AND (“screw*”)). Language or publication year was not restricted. Further, relevant articles and their bibliographies were manually searched after the initial electronic search.

### 2.2. Study Selection

In this systematic review, the following inclusion criteria were used: (1) study: directly compared the DHS and CCS techniques in FNF (double-arm study); (2) population: patients diagnosed with Pauwels type II or type III FNF; (3) intervention: DHS fixation using the conventional DHS system only, not other fixed-angle systems; (4) control: CCS fixation; (5) outcomes: nonunion and ONFH. We excluded studies that (1) reported treatment of childhood FNF and (2) did not report treatment for traumatic FNF, such as pathological fractures. Non-original articles including biomechanical or cadaveric studies, technical notes, letters to the editor, conference abstracts, expert opinions, review articles, meta-analyses, and case reports were excluded; only original research was included.

After duplicate results were removed, two board-certified orthopedic surgeons (L.E.J., K.C.-H.) who had completed an orthopedic hip and pelvic trauma fellowship independently selected the studies for full-text review on the basis of the titles and abstracts of the papers. If data presented in the abstract were insufficient for decision making, the full article was reviewed. Attention was paid to the outcomes including nonunion or ONFH in DHS and CCS groups, respectively, for meta-analysis. During the screening process, discussion between the two researchers resolved any ambiguous situation or bias about the eligibility of the papers.

At each stage of the literature search, we calculated kappa values to determine inter-reviewer agreement for the study selection. Agreement between reviewers was correlated with kappa values as follows: κ = 1 indicated “perfect” agreement, 1.0 > κ ≥ 0.8 indicated “almost perfect” agreement, 0.8 > κ ≥ 0.6 indicated “substantial” agreement, 0.6 > κ ≥ 0.4 indicated “moderate” agreement, 0.4 > κ ≥ 0.2 indicated “fair” agreement, and κ < 0.2 indicated “slight” agreement.

### 2.3. Data Extraction

For qualitative data synthesis, we used a standardized approach to extract the following information and variables from the selected studies: (1) study design; (2) number of patients; (3) mean age; (4) sex; (5) Pauwels classification type; (6) mode of injury; (7) initial displacement; (8) timing of surgery from injury; (9) operation time; (10) nonunion; (11) ONFH; (12) postoperative infection; (13) mean follow-up period.

Because our meta-analysis was designed to consider only variables that can be extracted from data reported in more than three research articles, we extracted data on postoperative nonunion and ONFH for the pooled analysis.

For the meta-analysis, if the required data were not reported in the article, we attempted to calculate them from the full-text review, and if the data still could not be obtained, the study authors were contacted. Throughout the data extraction process, two investigators independently extracted data and resolved any disagreements through discussion.

### 2.4. Quality Assessment

We used the Downs and Black scale to assess the quality of included studies [10]. This scale includes reporting, external validity, internal validity, and power. Two researchers (L.E.J., K.C.-H) independently assessed the quality of each study. Then, total scores and interobserver agreement was calculated.

### 2.5. Risk of Bias Assessment

Methodological was assessed using the risk of bias in nonrandomized studies of interventions (ROBINS-I) scoring system [11], which is a valid tool for assessing the qualities of nonrandomized studies. ROBINS-I covers seven domains: confounding, selection, intervention classification, deviation from intervention, missing data, measurement of outcome, and selection of reported result. Each domain is graded as low, moderate, serious, and critical.

### 2.6. Data Synthesis and Statistical Analysis

The main outcomes of the present meta-analysis were a comparison of postoperative nonunion rate and incidence of ONFH. For outcome comparisons, odds ratios (ORs) and 95% confidence intervals (CIs) were calculated as dichotomous data. Heterogeneity was assessed using the I^2^ statistic, and 25%, 50%, and 75% were considered low heterogeneity, moderate heterogeneity, and high heterogeneity, respectively. The outcomes, pooled estimate of effects, and overall summary effect of each study were presented using forest plots. Statistical significance was set at *p* < 0.05. We pooled all data using a random-effect model that has previously been recommended to avoid overestimation of study results especially in medicine [12]. We did not perform test for publication bias as, in accordance with the Cochrane guidelines, it is typically only recommended when at least 10 studies are included in the meta-analysis [13]. All statistical analyses were performed using Review Manager (RevMan) version 5.3.

## 3. Results

### 3.1. Study Identification

The details of the study identification and selection process are summarized in Figure 1. The initial electronic literature search yielded 1727 articles. After removing 872 duplicates and adding four publications that were identified by manual searching, 822 studies were screened. Of the 822 studies, 793 were excluded after the titles and abstracts were screened, and 24 studies were excluded after full-text review because 20 of them were not in the field of interest and data reported in the other four studies were insufficient for comparative study. Thus, five studies were eligible for data extraction and meta-analysis. The agreement on study selection between the reviewers at the title review and abstract review stages was significant (κ = 0.785 and 0.782, respectively). At the full-text review stage, the interobserver agreement was perfect (κ = 1.0).

### 3.2. Study Characteristics and Qualitative Synthesis

Of the five studies [5,14,15,16,17], four [14,15,16,17] were retrospective cohort studies and one [5] was a randomized controlled trial. In total, there were 252 patients with FNF. The DHS technique was used when treating 96 of the patients (DHS group), whereas the CCS technique was used when treating 156 of the 252 patients (CCS group). The mean patient age ranged from 28.8 to 47.7 years. The proportion of male patients ranged from 49.0% to 79.1%. All five studies reported Pauwels type III FNF, whereas two studies [5,14] reported Pauwels type II and type III FNFs. The mean follow-up period ranged from 10.5 to 30.0 months. More details on each included study (including the measured outcome of nonunion and ONFH) are presented in Table 1.

Two studies described the mode of injury [14,15]. Nine of the 115 patients (8%) were injured because of a fall from standing height, and 106 of the 115 patients (92%) were injured because of a traffic accident or a fall from heights. Three studies presented an initial displacement of FNF [15,16,17]. One study demonstrated that all the cases were displaced [15]. Two studies stated that 14 of the 129 cases (11%) were nondisplaced FNF and that 15 of the 129 cases (89%) were displaced FNF. Surgery was performed at <24 h from injury in two studies [14,16], and three studies demonstrated that the mean timing of surgery from injury was >24 h [5,15,17]. The operation time was measured in two studies [14,15]. Postoperative infection was described in three studies [5,15,16], and two cases presented with postoperative infection in the DHS group. More details are given in Table 2.

### 3.3. Quality Assessment

The included studies score between 13 and 16 points for quality (Table 3). All of the studies clearly described objective, patient characteristics, intervention, outcome, and statistical test. No study provided the data of the total population or the sample population, resulting in a limitation for the assessment of the representativeness of the included patients (3-point deduction). Because four included studies were retrospective studies, there were point deductions due to randomization (2 points), blindness (2 points), concealment (1 point), and confounding (1 point). In addition, “characteristics of patients loss to follow up” and “adjustment for differing lengths of follow-up” made 2-point deductions in all study.

### 3.4. Risk of Bias Assessment

The risk of bias was moderate in three studies [14,15,16] and serious in two studies [5,17] (Table 4). Most of the studies were nonrandomized, retrospective studies, and subject to the biases of confounding, selection, and intervention classification. The bias due to confounding variables was moderate in two studies, because intervention and control groups presented different quality of reduction [5,16]. A serious selection bias was found in two included studies where different indications for intervention and control groups were described [5,17]. The bias due to measurement of outcome and selection of reported results were low in all included studies.

### 3.5. Meta-Analysis Results

#### 3.5.1. Nonunion Rate

Data on nonunion rate following the use of the DHS and CCS fixation techniques for Pauwels type II or type III FNF were extracted from all five included studies. There were 3 and 21 cases of nonunion in the DHS and CCS groups, respectively. The pooled analysis revealed that nonunion rate was significantly higher in the CCS group than in the DHS group (OR = 0.32; 95% CI, 0.11–0.96; *p* = 0.04). The heterogeneity was considered low (I^2^ = 0%), and the forest plot and details are shown in Figure 2.

#### 3.5.2. Incidence of ONFH

Four of the five studies [5,14,16,17] included data on incidence of ONFH in the DHS and CCS groups. All four studies compared the incidence of ONFH between the two groups treated for FNF. ONFH was reported in nine out of a total of 72 patients in the DHS group and in 17 out of a total of 133 patients in the CCS group. The pooled analysis showed no statistically significant differences in the incidence of ONFH between the two groups (OR = 0.98; 95% CI, 0.20–4.73; *p* = 0.98). The heterogeneity was considered moderate (I^2^ = 53%), and the forest plot and details are shown in Figure 3.

## 4. Discussion

The principal finding of this pooled analysis was that the nonunion rate was higher in the CCS group than in the DHS group for Pauwels type II or type III FNF, and there was no difference in the incidence of postoperative ONFH between the two groups.

Although it was excluded from this meta-analysis at the final full-text review stage because we could not extract enough data for pooling, the study by Lee et al. [18] compared the DHS and CCS techniques retrospectively in 90 patients with undisplaced FNF. In the study, it was reported that there was no mechanical failure or nonunion in the DHS group, but a 9.4% implant failure rate and a 3.1% nonunion rate in the CCS group were reported without statistical significance. In contrast, ONFH was observed in 12% and 9.4% of the DHS and CCS groups, respectively. These results are comparable with the results of our pooled analysis.

Several earlier studies have compared nonunion rate between DHS and CCS groups of patients with FNF. In their retrospective cohort study of 86 consecutive patients with FNF that was published in 2017, Chen et al. reported no difference in nonunion rate between the DHS and CCS groups [19]. In their more recent prospective analysis of 54 patients with FNF, Shu et al. reported nonunion rates of 7.1% and 7.7% in the DHS and CCS groups, respectively, and insisted that the two fixation techniques may have equal effectiveness in terms of fracture union [20]. In addition, two systematic reviews compared complications in FNF between the DHS and CCS techniques [21,22]. Both studies concluded that no significance difference in nonunion existed between the DHS and CCS techniques.

However, closer analysis revealed that these earlier studies did not consider the verticality of the fracture line in their analysis. In contrast, we included only Pauwels type II and III, which is the differentiation of the present study. In the treatment of high-shear-angle FNFs, tilting of the head fragment by a vertical shearing force should be prevented with the screws that anchor to the trochanteric portion. In clinical situations where partially threaded screws are used, the possibility of insufficient stability at the trochanteric portion is high. In terms of biomechanics, it has been shown that one DHS device is stronger than three parallel CCS devices in the treatment of basicervical fractures, the orientation of which is similar but distal to that of Pauwels type III fractures [2,23]. Further, in a recent clinical study of 78 patients with FNF conducted by Sahin et al., nonunion rates of 12% and 21% were reported in the DHS and CCS groups, respectively [24]. On the basis of the results of our pooled analysis, the risk of fracture nonunion is higher with the CCS fixation technique than with the DHS fixation technique, especially in patients with vertically oriented FNF.

In our study, we did not observe differences in the incidence of ONFH between the DHS and CCS groups. The incidence of ONFH following FNF has been reported to range from 10% to 45% [16]. In an earlier study, it was reported that the two most important factors in the development of ONFH in young active patients are fracture displacement and quality of reduction [25]. However, only a limited number of studies have evaluated the incidence of ONFH after different FNF fixation methods with consideration for the fracture displacement. Three of the five included studies presented that most of the cases (115/129, 89%) were displaced [15,16,17], but two studies did not present any information about displacement [5,14]. However, with relatively young patients of mean ages ranging from 28.8 to 47.7 years in the present meta-analysis, we could presume that the nondisplaced FNF would be small, and we performed meta-analysis without considering the initial displacement. On the basis of the results of our pooled analysis, we are of the opinion that the verticality of the fracture pattern has a stronger effect on the development of ONFH than implant choice. Furthermore, it is necessary to conduct further high-quality prospective studies that simultaneously consider fixation techniques and fracture patterns in patients with FNF who have ONFH following treatment.

Several meta-analyses have reported that the surgical timing affects union but not ONFH [26,27]. It follows that the surgical factors may have a stronger effect on the nonunion rate, whereas the natural course, according to the fracture pattern, may have a stronger effect on the incidence of ONFH than on the implant choice. In the present study, a wide range of timing of surgery from injury and subanalysis of nonunion according to the surgical timing seemed meaningful, but it was not performed due to data insufficiency. Regarding ONFH, it can be assumed that the correlation between surgical timing and ONFH was not significant considering the literature published so far. However, previous studies cannot guarantee the delay in surgery even in terms of ONFH. Most of the previous meta-analyses were based on retrospective studies, and we anecdotally experienced poor outcomes resulting from delayed surgery. Further research is thus needed to prove the relationship between the timing of fixation and ONFH.

Iatrogenic rotation of head fragment by hip screws is one of the disadvantages of DHS fixation, especially in young patients. Antirotation screws are used to avoid this problem, and they also provide additional stability [28]. Three of the five included studies [14,15,17] reported the use of antirotation screws. Recently, a femoral neck system proven to biomechanically provide stability comparable with that of a DHS system has been used [29]. Without the iatrogenic rotational forces, this implant can be a valid alternative for the treatment of high-shear-angle FNFs. However, because of the recent use and lack of studies on it, the femoral neck system was excluded from our pooled analysis. We consider it necessary that further evaluation in the future is conducted using sufficient data.

This study has several limitations. First, the number of included studies is relatively small. Even after a systematic search with no restrictions on language and publication year, we identified only five suitable studies for quantitative synthesis. Nevertheless, a meta-analysis is appropriate for the generation of a higher level of evidence in studies for which large cohorts are not feasible. Second, except for one study, all the included studies were retrospective in nature. Pooling results based on predominantly retrospective studies can lead to an overestimation of outcomes. In addition, the indication of DHS or CCS was not described specifically, which could affect selection bias. Third, due to limited available data, we could only conduct meta-analysis for postoperative nonunion rate and incidence of ONFH. Fourth, the mode of injury, initial displacement, reduction method, and reduction state were not controlled in our analysis. In retrospect, implant choice was likely informed by these factors, and there is the possibility of selection bias. Further, as they are known to be associated with nonunion and ONFH, these factors may have influenced our results. Therefore, further high-quality studies that consider the various clinical outcomes and complications of the DHS and CCS fixation techniques in the treatment of vertically oriented FNFs are needed.

## 5. Conclusions

The DHS fixation technique is more favorable than the CCS fixation technique for the treatment of vertically oriented FNFs, especially as it is associated with a lower risk of fracture nonunion.

## Figures and Tables

**Figure 1 jpm-11-01017-f001:**
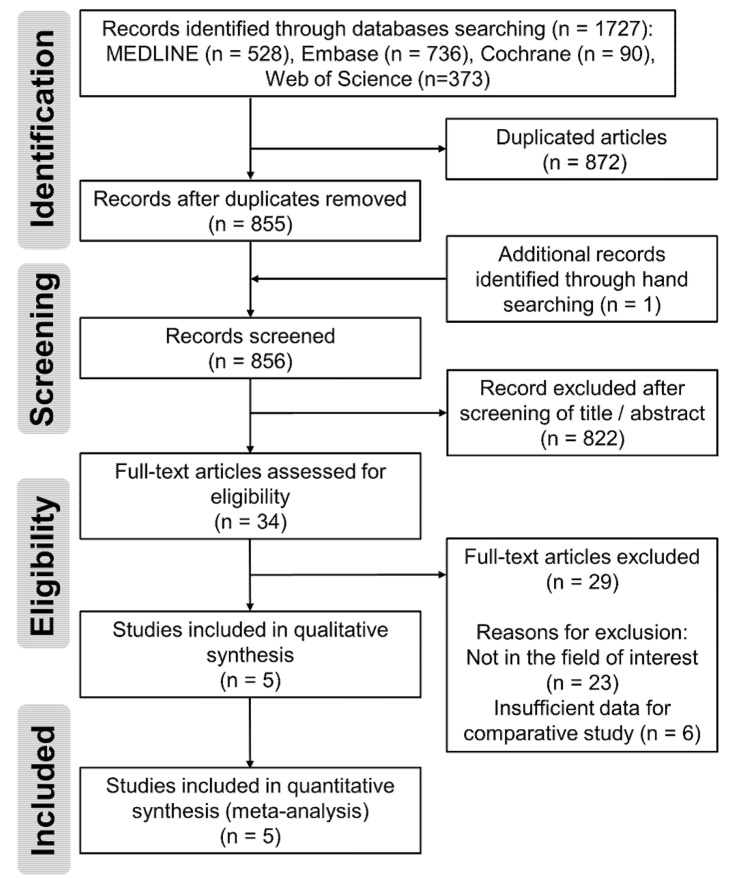
PRISMA (Preferred Reporting Items for Systematic Reviews and Meta-analyses) flow diagram of literature selection.

**Figure 2 jpm-11-01017-f002:**
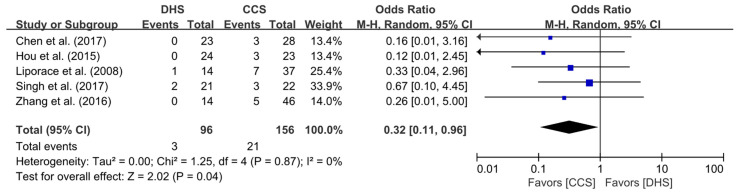
Results of aggregate analysis for comparison of nonunion rate according to fixation techniques.

**Figure 3 jpm-11-01017-f003:**
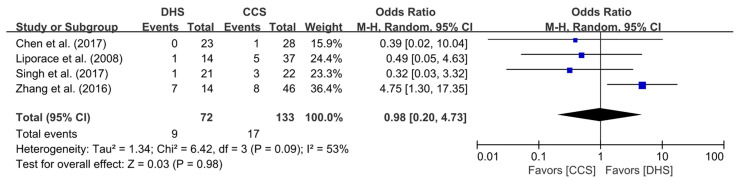
Results of aggregate analysis for comparison of incidence of ONFH according to fixation techniques.

**Table 1 jpm-11-01017-t001:** Study design, demographic data, and incidence of NU and ONFH.

Author (Year)	Study Design	Number of Patients		Mean Age, Years	Male Sex, %	Pauwels Type	Mean Follow-Up Period, Months	NU		ONFH	
		DHS	CCS					DHS	CCS	DHS	CCS
Chen et al. (2011) [14]	RCS	23	28	41.8	49.0	II, III (41.2%:58.8%)	≥12	0 (0%)	3 (11%)	0 (0%)	1 (4%)
Hou et al. (2015) [15]	RCS	24	23	43.4	55.3	III	30.0	0 (0%)	3 (13%)	NA	NA
Liporace et al. (2008) [16]	RCS	14	37	42	NA	III	24	1 (7%)	7 (19%)	1 (7%)	5 (14%)
Singh et al. (2017) [5]	RCT	21	22	28.8	79.1	II, III (53.5%:46.5%)	10.5	2 (10%)	3 (14%)	1 (5%)	3 (14%)
Zhang et al. (2016) [17]	RCS	14	46	47.7	73.1	III	21.6	0 (0%)	5 (11%)	7 (50%)	8 (17%)

Abbreviations: CCS, cannulated cancellous screw; DHS, dynamic hip screw; NA, not applicable; NU, nonunion; ONFH, osteonecrosis of femoral head; RCS, retrospective comparative study; RCT, randomized controlled trial.

**Table 2 jpm-11-01017-t002:** Details of injury characteristics, operation time, and postoperative infection.

	Mode of Injury		Initial Displacement	Timing of Surgery from Injury		Operation Time (min)		Postoperative Infection	
	DHS	CCS		DHS	CCS	DHS	CCS	DHS	CCS
Chen et al. (2011) [14]	Low energy 1 (4%) High energy 22 (96%)	Low energy 1 (4%) High energy 27 (96%)	NA	9.2 (2–18) h	9.0 (2–16) h	48.3 ± 5.3	44.0 ± 3.6	NA	NA
Hou et al. (2015) [15]	Low energy 2 (7%) High energy 28 (93%)	Low energy 5 (15%) High energy 29 (85%)	Displaced 47 (100%)	32.0 (2–72) h	33.0 (3–67) h	51.0 ± 8.7	49.0 ± 8.3	0 (0%)	0 (0%)
Liporace et al. (2008) [16]	NA	NA	Displaced 58 (91%) Nondisplaced 4 (9%)	<24 h		NA	NA	1 (7%)	0 (0%)
Singh et al. (2017) [5]	NA	NA	NA	6.2 days		NA	NA	1 (5%)	0 (0%)
Zhang et al. (2016) [17]	NA	NA	Displaced 57 (85%) Nondisplaced 10 (15%)	2.3 ± 0.8 days		NA	NA	NA	NA

Low energy means fall from standing height, and high energy includes traffic accident and fall from height. Abbreviations: CCS, cannulated cancellous screw; DHS, dynamic hip screw; IF, internal fixation; NA, not applicable.

**Table 3 jpm-11-01017-t003:** Downs and Black score for quality assessment.

	Downs and Black Total Score		Crude Agreement	Cohen’s Kappa Coefficient
	Reviewer 1	Reviewer 2		
Chen et al. (2011) [14]	13	14	95.8%	0.92
Hou et al. (2015) [15]	16	16	91.7%	0.81
Liporace et al. (2008) [16]	16	15	95.8%	0.91
Singh et al. (2017) [5]	13	16	87.5%	0.74
Zhang et al. (2016) [17]	12	13	95.8%	0.92

Two reviewers (L.E.J. and K.C.-H.) performed quality assessment separately.

**Table 4 jpm-11-01017-t004:** The risk of bias assessment of included studies using ROBINS-I tool.

Study	Confounding	Selection	Intervention Classification	Deviation from Intervention	Missing Data	Measurement of Outcome	Selection of Reported Result	Overall
Chen et al. (2011) [14]	Low	Low	Moderate	Low	NI	Low	Low	Moderate
Hou et al. (2015) [15]	Low	Moderate	Moderate	Low	NI	Low	Low	Moderate
Liporace et al. (2008) [16]	Moderate	Low	Moderate	Low	Low	Low	Low	Moderate
Singh et al. (2017) [5]	Moderate	Serious	Low	Moderate	Low	Low	Low	Serious
Zhang et al. (2016) [17]	Low	Serious	Moderate	Moderate	Low	Low	Low	Serious

Abbreviation: NI, no information.

## Data Availability

Not applicable.

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
