# Peer review of "Dynamic Hip Screw versus Cannulated Cancellous Screw in Pauwels Type II or Type III Femoral Neck Fracture: A Systematic Review and Meta-Analysis"

_jpm, 2021, doi:10.3390/jpm11101017_

Round 1
Reviewer 1 Report
The authors did Systematic Review and Meta-analysis titled: Dynamic Hip Screw versus Cancellous Cannulated Screw in Pauwels Type II or Type III Femoral Neck Fracture.
The paper is good but has some methodological fallbacks that should be repaired:
- Inclusion criteria should be presented more precisely (please provide inclusion criteria in regards to: participants, intervention, comparison and outcomes).
- Please provide initials of authors who selected the studies for full-text review. Also, it should be stated how disagreements among the authors were resolved.
- The authors searched three databases. At least four databases need to be explored for an efficient search in reviews. (REFERENCE: Bramer WM, Rethlefsen ML, Kleijnen J, Franco OH. Optimal database combinations for literature searches in systematic reviews: a prospective exploratory study Syst Rev. 2017, 6, 245.).
- Downs and Black scale scores for the included studies by author 1 and author 2 should be presented in a separate Table. The total scores and inter-observer agreement should be depicted in the table, as well.
Reviewer 2 Report
Thank you for the excellent paper. This is the first we aimed to compare the DHS and CCS fixation techniques in Pauwel type II or type III FNFs, with special focus on nonunion rate and 55 incidence of postoperative ONFH.
Major bias
: Fracture type and displacement is important to assess instability.
Pauwel type II or type III is only fracture angle. Pauwel type II with displacement VS Pauwel type III without displacement was included? If possible, the combination Garden type and pauwel type is the option for fracture morphology.
: as you pointed out, ON is dependent on the fracture displacement, surgical timing, and follow up period.
You did not adjusted these factor. It is limitation due to multifactorial reason for outcome on observational study.
Minor revision
Method: Did you register your study protocol? Reporting bias?
Method: Pauwels Type II is different from Type III, please add results of subgroup analysis per each fracture type if possible.
Method : why did you select the all paper which reported NU or ONFH ? only reported both outcomes?
Method : did you include poor quality of studies?
Method : why did you select MINORS not Robins 1 tool?
https://methods.cochrane.org/methods-cochrane/robins-i-tool
Figure 2: please add the favour direction on horizontal axis. For example, unfavour CCS
Discussion : please write your strength, and clinical implication clearly and shortly based on your results compared to previous studies.
Round 2
Reviewer 1 Report
The authors did everything as asked, being that this is not a Cochrane nor its rules I would say the authors did a good job.
Reviewer 2 Report
Thank you for revision.
Major:
Please reflect these comments in the revision manuscript. I can not understand which sentence is corresponding to my requests, with academic writing.
Minnor;
You could resister your study protocol at the other site, for example the here: https://www.protocols.io/
Therefore, your study have Reporting bias.
4: We did not present this result in the revised manuscript, because this finding cannot provide meaningful finding.
→It is reporting bias.
